# Fractional Moore-Gibson-Thompson heat transfer model with nonlocal and nonsingular kernels of a rotating viscoelastic annular cylinder with changeable thermal properties

**Ahmed E. Abouelregal** [1,2,3]*, **Meshari Alesemi**[4]

**1** Department of Mathematics, College of Science and Arts, Jouf University, Al-Qurayyat, Saudi Arabia,
**2** Basic Sciences Research Unit, Jouf University, Sakaka, Saudi Arabia, **3** Department of Mathematics,
Faculty of Science, Mansoura University, Mansoura, Egypt, **4** Department of Mathematics, College of
Science, University of Bisha, Bisha, Saudi Arabia

* ahabogal@gmail.com

Fractional Moore-Gibson-Thompson heat transfer
model with nonlocal and nonsingular kernels of a
rotating viscoelastic annular cylinder with
changeable thermal properties. PLoS ONE 17(6):
e0269862. https://doi.org/10.1371/journal.
pone.0269862

University, CHINA

**Data Availability Statement:** All relevant data are
within the manuscript.

## Abstract

Long hollow cylinders are commonly utilized in various technological applications, including
liquid and gas transmission. As a result, its value is growing, becoming increasingly impor-
tant to many research efforts. Compared with thermal isotropic homogeneous cylinders,
thermo-viscoelastic orthotropic cylinders have less relevant data. In this paper, a thermoe-
lastic fractional heat conduction model was developed based on the Moore-Gibson-Thomp-
son equation to examine the axial symmetry problem of a viscoelastic orthotropic hollow
cylinder. Atangana and Baleanu derivative operators with nonsingular and nonlocal kernels
were used in constructing the fractional model. The thermal properties of the cylinder materi-
als are assumed to be temperature-dependent. The Laplace transform is applied to solve
the system of governing equations. The numerical calculations for temperature, displace-
ment, and stress components are performed by the effect of fractional order, rotation, and
changing thermal properties of the cylinder. The results showed that due to the presence of
fractional derivatives, some properties of the physical fields of the medium change accord-
ing to the value of the fractional order.

## 1 Introduction

The investigation of viscoelastic behaviour is useful in several situations. For instance, materi-
als utilized in real structural applications may display viscoelastic behaviour, which signifi-
cantly impacts the material's efficiency [1]. Viscoelastic activity can occur as an unexpected
side effect of materials utilized in structural applications. In some applications, the viscoelastic-
ity of many of these materials may be intentionally used in the system design to achieve a spe-
cific aim. Furthermore, the applied mathematics community is interested in the mathematics
that underpins viscoelasticity theory. Moreover, because viscoelasticity is physically connected
to a number of microphysical mechanisms and may be utilized as an experimental probe of
those interactions, it is of importance in various disciplines of materials science, heavy

**Funding:** Ahmed E Abouelregal Jouf University DSR-2021-03-0377 The funders had no role in study design, data collection and analysis, decision to publish, or preparation of the manuscript.

**Competing interests:** No

**Abbreviations:** $\beta_{ij}$, coupling coefficients; $\alpha_t$, thermal expansion coefficient; $C_E$, specific heat; $T_0$, environmental temperataure; $\theta = T-T_0$, temperature increment; T, absolute temperature; $F_i$, $e$body forces components; $\alpha$, fractional-order; $e = e_{kk}$, cubical dilatation; $\sigma_{ij}$, stress tensor; $e_{ij}$, strain tensor; $q_i$, heat flux components; $K_{ij}$, thermal conductivity tensor; $\rho$, material density; Q, heat source; t, the time; $K_{ij}*$, thermal conductivity rates; $\nabla^2$, laplace operator; $\Omega$, angular velocity; $u_i$, displacement components; $t_0$, thermal relaxation time; $C_{ijkl}$, elastic constants; $\eta$, viscoelastic relaxation time; $\vartheta=\theta$, thermal displacement.

engineering, and solid-state physics. Finally, when using viscoelastic testing as an instrument, the causal linkages between viscoelasticity and microstructure are explored [2].

Thermoelasticity, which admits a limited speed for the transmission of thermal signals, has received a lot of attention in the last two decades. Generalized theories differ from conventional concepts [3] in that they are based on a parabolic-type heat equation. Several writers, for various reasons, have proposed these generalized ideas. Green and Lindsay [4] created a theory by incorporating the temperature rate among the constitutive variables. In contrast, Lord and Shulman [5] established an approach based on a modified heat transfer law that includes a heat-flux rate. The heat equation for this theory is the hyperbolic equation, which asserts that heat and elastic waves travel at a controlled speed.

The Green–Naghdi thermoelastic theory [6–8] has been developed to give a coherent idea that explains both the elastic and heat waves associated with the second sound. Heat pulse propagation can be achieved using the Green–Naghdi approach based on rational thermodynamics. In this theory, theoretical types I, II, and III are divided into three models. A variety of heat flow concerns can be represented in this theory. Their technique is unusual because it includes a field called thermal displacement, whose time derivative matches the empirical temperature. The third type is a general extension that may be used to explain a variety of things, including the usual Fourier idea and undamped thermal wave propagation.

Chandrasekharaiah [9, 10] proved unique theorems by employing the energy approach. Many attempts have been made to examine various theoretical and practical aspects of thermoelasticity under the Green-Naghdi Type II or Type III models. Choudhuri proposed the Green–Naghdi-inspired three-phase lag thermoelasticity theory [11]. Abouelregal [12–17] tried to alter the basic Fourier law by using higher-order temporal derivatives.

Unfortunately, the Green-Naghdi Type III (GN-III) equation, like the Fourier equation, has the flaw that thermal waves spread quickly [18, 19]. This mechanism did not follow the causality principle. A feasible solution to this problem is replacing the GN-III model's constitutive equation with a relaxation time coefficient. Conti et al. [19] and Quintanilla [20, 21] corrected the basic law in the GN-III model using the Maxwell and Cattaneo approach. Quintanilla in [20] is working on a novel thermoelastic Moore–Gibson–Thompson model. Quintanilla [20] improved the Green-Naghdi model of type III by including the relaxation parameter. Many scholars are interested in the Moore–Gibson–Thompson Equation (MGT), and various publications have been written about it. A third-order differential equation was used to build this concept, which is relevant in many fluid dynamics difficulties [21]. Since its start, papers devoted to the Moore-Gibson-Thomson theory have risen dramatically [22–31].

Complicated problems are explained using fractional-order derivatives and partial differential equations (PDEs). One of the difficulties encountered in solving such equations is predicting the future behaviour of the physical situation. Investigators can use fractional derivative operators to help them solve this dilemma. Due to the ability of non-integer order derivatives to examine the complex behaviour of several phenomena, such as heredity and memory qualities of materials and structures, fractional calculus (FC) modelling of dynamical settings is becoming increasingly popular these days [32]. Non-integer order derivative formulae can also accurately correlate to actual results [33]. Consequently, FC urged scholars and practitioners to continue working on fractional calculus to characterize dynamical systems correctly.

Fractional derivatives have been defined in several ways in the literature. The Riemann–Liouville and Caputo derivative operators [34] are the most commonly used fractional derivative operators in dynamic model issues. However, they only have a single kernel. In 2016, Atangana and Baleanu [35] proposed new formulations of Caputo and Fabrizio's non-integer order operator with a nonsingular kernel, first proposed in 2015. The kernels of these new derivative operators are smooth, and they display all of the characteristics of Caputo and

Riemann–Liouville operators. For temporal and geographic variables, the Caputo and Fabrizio derivative operator is adequate, while in material and thermal sciences, the Atangana and Baleanu derivative operator, given in terms of the Mittag-Leffler function [36], is advantageous. Atangana et al. [37] offer a thorough investigation, including numerical results, stability evaluations, and error analyses. Because of the Sumudu transformation, Atangana and Akgul [38] sought to build new transfer functions that would add considerably to the new graphs of Bode, Nichols, and Nyquist. During the last several years, the primary characteristics of these operators have been extensively investigated for a variety of real applications [39–54].

A significant proportion of the study has been done using non-temperature-dependent material characteristics, limiting the solutions' application to certain temperature ranges. The physical qualities of modern structural parts are often subjected to such huge temperature variations that they can no longer be termed constant, even in a broad sense [55]. At high temperatures, the coefficients of elasticity and thermal conductivity of linear thermal expansion of materials are no longer constants. As a result, because the thermal and mechanical properties of materials change with temperature [56], the temperature-dependent characteristics of these materials must be considered in the study of thermal stresses for these materials. Due to high temperatures, high gradient temperatures, and cyclical temperature variations, thermal stresses are applied to structural components and mechanical elements in nuclear power plants, chemical plants, and high-speed aircraft [57]. The structures mentioned above, components, and materials' thermomechanical behaviour have become increasingly important. The degree to which temperature influences material quality changes as the temperature rises. Temperature-dependent material characteristics significantly influence thermal stress at high temperatures or at high gradient temperatures [57].

The main purpose of the research is to reformulate and apply a thermo-viscoelastic heat transport model to rotating temperature-dependent materials. The proposed heat transport model uses the Moore-Gibson-Thompson (MGT) equation. In addition to the Caputo fractional derivative, the Atangana and Baleanu fractional derivative operators with nonsingular and nonlocal kernels are discussed. Researchers use the extended thermoelastic MGT model to investigate the thermoelastic issue of an infinite body with a cylindrical cavity and temperature-dependent material characteristics.

The governing equations were constructed to be nonlinear because the properties are temperature-dependent. Due to the nonlinearity of the governing equations, an appropriate mapping is needed to convert the heat equation to a linear equation. The Laplace transform is used to process fundamental equations based on numerical, analytical approaches. According to the findings, temperature-dependent aspects diminish the magnitudes of the physical variables evaluated. This shows that considering the temperature dependency of features in generalized thermoelastic settings is vital and practical for accurately anticipating thermoelastic behaviour.

## 2 Governing equations for fractional MGT thermo-viscoelastic model

The Kelvin-Voigt model is a micromechanical model commonly used to describe the behaviour of viscoelastic matter. When the deformation is time-dependent but recoverable, the model depicts the delayed elastic stress response. The governing equations for a homogeneous generalized thermo-viscoelastic material can be written as follows, according to Abouelregal [23, 24], Green–Naghdi [7], and Lord and Shulman [5]:

The stress-displacement-temperature relation

$$\sigma_{ij} = \tau_m c_{ijkl} e_{kl} - \tau_m \beta_{ij} \theta, \tag{1}$$

where $\tau_m = 1 + \eta \frac{\partial}{\partial t}$ and $\eta$ is the viscoelastic relaxation time due to the viscosity.

The strain-displacement relation

$$2e_{ij} = u_{j,i} + u_{i,j} \tag{2}$$

The equation of motion

$$\sigma_{ij,j} + F_i = \rho \frac{\partial^2 u_i}{\partial t^2}. \tag{3}$$

Consider a homogeneous generalized thermoelastic solid rotating with a uniform angular velocity $\Omega = \Omega\mathbf{n}$, where $\mathbf{n}$ is a unit vector defining the rotation axis. As a result of the rotation, the equation of motion now includes two additional components: the centripetal force $(\Omega \times (\Omega \times \mathbf{u}))$ due to time-varying motion only and the acceleration of Coriolis $(2\Omega \times \dot{\boldsymbol{u}})$ due to the moving reference frame. This indicates that the motion Eq (3) as a result of rotation takes the following form:

$$\rho \frac{\partial^2 u_i}{\partial t^2} + \rho(\boldsymbol{\Omega} \times (\boldsymbol{\Omega} \times \mathbf{u}))_{\boldsymbol{i}} + \rho(2\boldsymbol{\Omega} \times \dot{\boldsymbol{u}})_i = \sigma_{ij,j} + F_i. \tag{4}$$

The energy balance equation is given by

$$\rho C_E \frac{\partial \theta}{\partial t} + T_0 \tau_m \frac{\partial}{\partial t}\left(\beta_{ij} e_{ij}\right) = -q_{i,i} + Q. \tag{5}$$

Cattaneo-Vernotte created a modified version of Fourier's law in the following form by including the relaxation-time parameter concerning the heat flow vector

$$\left(1 + \tau_0 \frac{\partial}{\partial t}\right) q_i = -K_{ij}\theta_{,i}. \tag{6}$$

According to the GN-III model [7], the modified Fourier law is as follows:

$$q_i = -K_{ij}\theta_{,i} - K_{ij}^*\vartheta_{,i}. \tag{7}$$

The modified MGT non-Fourier law heat equation is given by [20, 21, 58]

$$\left(1 + \tau_0 \frac{\partial}{\partial t}\right) q_i = -K_{ij}\theta_{,i} - K_{ij}^*\vartheta_{,i}. \tag{8}$$

The concepts of a fractional-order derivative and a partial differential equation are utilized to describe difficult situations. Predicting the future behaviour of a physical problem is one of the difficulties in solving such equations. Many definitions of partial derivatives may be found in the literature. The Caputo model of the captive derivative [48] is often utilized to mimic real-world issues since it asserts that initial circumstances are taken into account. This variable, however, presents a singularity problem due to the function employed to generate the local derivative. In [59], Caputo and Fabrizio introduce a new definition of fractional derivation. Atangana and Baleanu [35] introduced a novel fractional derivative type with nonsingular kernels, including the Mittag-Leffler function. The Atangana–Baleanu (AB) derivative is a novel formulation for dynamical systems with memory impact that gives a better explanation. The kernels of these operators are nonlocal and nonsingular.

The Riemann-Liouville fractional integral (fractional-order derivative) is used in the standard fractional-order constitutive model, which is described as [59]

$$D_t^\alpha f(t) = \frac{1}{\Gamma(1-\alpha)} \frac{d}{dt} \int_0^t \frac{f(\xi)}{(t-\xi)^\alpha} d\xi, 0 < \alpha \leq 1. \tag{9}$$

The underlying ideas are the Riemann-Liouville and Caputo conceptions, which are concerned with the solitary kernel $K_e(t,\xi) = \frac{(t-\xi)^{-\alpha}}{\Gamma(1-\alpha)}, 0 < \alpha \leq 1$.

In this paper, the Atangana–Baleanu fractional operator of order $\alpha$ is used to simulate the time-fractional MGT thermoelastic heat conduction model. Eq (4) may be expressed as in this situation

$$(1 + \tau_0 D_t^{(\alpha)}) q_i = -K_{ij}\, \theta_{,i} - K_{ij}^* \vartheta_{,i}. \tag{10}$$

Based on the Caputo sense, the new fractional derivative of Atangana–Baleanu $D_t^{(\alpha)}$ of order $\alpha \in (0,1)$ is given by [35]

$$D_t^{(\alpha)} f(t) = \frac{1}{1-\alpha} \int_0^t \frac{\partial f(\xi)}{\partial \xi} E_\alpha \left[ -\frac{\alpha(t-\xi)^\alpha}{1-\alpha} \right] d\xi, \quad 0 < \alpha \leq 1, \tag{11}$$

where $E_\alpha(-t^\alpha) = \sum_{k=1}^\infty \frac{(-t)^{\alpha k}}{\Gamma(1+k\alpha)}$ denotes the generalized Mittag-Leffler function.

The significance of the Laplace transform approach in studying differential equations is well recognized. It is also recognized for $0 < \alpha \leq 1$ for this new fractional definition [35, 59]

$$\mathcal{L}[D_t^{(\alpha)} f(t)] = \frac{1}{1-\alpha} \frac{s^\alpha \mathcal{L}[f(t)] - s^{\alpha-1} f(0)}{s^\alpha + \frac{\alpha}{1-\alpha}}, \quad s > 0. \tag{12}$$

As a result, the Laplace transform will be beneficial when dealing with the Atangana–Baleanu fractional derivative. By substituting Eq (10) into Eq (5), the modified fractional thermoelastic model with the Atangana–Baleanu fractional derivative may be obtained:

$$(1 + \tau_0 D_t^{(\alpha)}) \left[ \frac{\partial}{\partial t} \left( \rho C_E \frac{\partial \theta}{\partial t} \right) + T_0 \tau_m \frac{\partial^2}{\partial t^2} (\beta_{ij} e_{ij}) - \frac{\partial Q}{\partial t} \right] = \frac{\partial}{\partial t} (K_{ij}\theta_{,i})_{,i} + (K_{ij}^*\theta_{,i})_i. \tag{13}$$

Eq (10) can be condensed to Quintanilla's proposed law [29] in the limited condition when the parameter $\alpha \to 1$. In much earlier thermoelastic and viscoelastic models, special instances may be generated from the preceding fractional heat Eq (13) in the following manner:

• The classical model of thermo-viscoelasticity (CTE) when $\tau_0 = K_{ij}^* = 0$.

• Lord and Shulman thermo-viscoelastic model (LS) (without fractional derivatives) when $K^* = 0$ and $\alpha = 1$.

• Lord and Shulman's theory with the Atangana–Baleanu fractional operator (FABLS) when we take $K^* = 0$ and $0 < \alpha < 1$).

• Type II of the Green and Naghdi models (without fractional derivatives) can be obtained when the terms, including the parameter $K_{ij}$ are ignored and $\tau_0 = 0$.

• Type III of the Green and Naghdi models (without fractional derivatives) can be acquired when the relaxation time $\tau_0 = 0$.

• The generalized Moore–Gibson–Thompson (MGTE) model of thermo-viscoelasticity (without fractional derivatives) is attained when $\tau_0$, $K^* > 0$ and $\alpha = 1$.

- The generalized Moore–Gibson–Thompson thermo-viscoelastic model with the Atangana–Baleanu fractional operator is attained when $\tau_0$, $K^* > 0$ and $0 < \alpha < 1$ (FABMGTE).

- Different models of thermoelasticity (without viscosity) can be obtained when the mechanical relaxation time $t_0$ is ignored.

## 3 Problem formulation

Orthotropic materials have material properties that change in three mutually orthogonal directions, each with a double rotational symmetry at a specific location. It is an anisotropic material whose properties vary depending on the angle it is viewed. Orthotropic materials include a wide variety of rolled crystals, polymers, and metals.

Cavity expansion models use the pressure solution for the constant expansion of a hole in an unbounded material as the contact pressure encountered by a projectile approaches a target. The problem under investigation is a rotating orthotropic thermo-viscoelastic body with a cylindrical cavity and a constant starting temperature of $T_0$. The cylindrical coordinate system $(r, \xi, z)$ is utilized with $z$-axis denoting the axial coordinate of the cylinder, where $R \leq r \leq \infty$, $0 \leq \phi \leq 2\pi$ and $0 \leq z \leq \infty$ are employed. The disturbances are assumed to be limited and contained at the $r = R$ border and disappear as $r \to \infty$. The Kelvin-Voigt linear viscoelasticity model can determine a material's viscoelastic characteristics.

The body cavity can be seen as a traction-free surface with a time-dependent thermal shock. Because the redial displacement $u_r = u(r,t)$ is simply the non-vanishing displacement component, the fundamental Eqs (1)–(3) and Moore–Gibson–Thompson heat transfer (MGTE) (13) without heat source ($Q = 0$) may be expressed as:

$$e_{rr} = \frac{\partial u}{\partial r}, \quad e_{\xi\xi} = \frac{u}{r}, \quad e_{r\xi} = e_{rz} = e_{z\xi} = 0, \tag{14}$$

$$(1 + \tau_0 D_t^{(\alpha)})\left[\frac{\partial}{\partial t}\left(\rho C_E \frac{\partial \theta}{\partial t}\right) + T_0 \tau_m \frac{\partial^2}{\partial t^2}\left(\beta_{11}\frac{\partial u}{\partial r} + \beta_{22}\frac{u}{r}\right)\right] = \frac{\partial}{\partial t}(\nabla.(K\nabla\theta)) + \nabla.(K^*\nabla\theta), \tag{15}$$

$$\begin{bmatrix} \sigma_{rr} \\ \sigma_{\xi\xi} \\ \sigma_{zz} \end{bmatrix} = \tau_m \begin{bmatrix} c_{11} & c_{12} & -\beta_{11} \\ c_{12} & c_{22} & -\beta_{22} \\ c_{13} & c_{23} & -\beta_{33} \end{bmatrix} \begin{bmatrix} \dfrac{\partial u}{\partial r} \\ \dfrac{u}{r} \\ \theta \end{bmatrix}, \tag{16}$$

Where $\sigma_{rr}$, $\sigma_{\xi\xi}$ and $\sigma_{zz}$ are the normal thermal stresses.

If we consider the rotation term about the $z$-axis to be a body force, the equation of motion in cylindrical coordinates is as follows:

$$\frac{\partial \sigma_{rr}}{\partial r} + \frac{\sigma_{rr} - \sigma_{\xi\xi}}{r} = \rho \frac{\partial^2 u}{\partial t^2} - \rho\Omega^2 u. \tag{17}$$

Using Eq (16), the previous equation of motion (17) is converted into the following form

$$\tau_m\left[c_{11}\left(\frac{\partial}{\partial r} + \frac{1}{r}\right)\left(\frac{\partial u}{\partial r}\right) - c_{22}\frac{u}{r^2}\right] = \tau_m \beta_{11}\frac{\partial\theta}{\partial r} + \tau_m(\beta_{11} - \beta_{22})\frac{\theta}{r} + \rho\frac{\partial^2 u}{\partial t^2} - \rho\Omega^2 u. \tag{18}$$

## 4 Boundary and initial conditions

To solve the system of equations, we will suppose that the medium mentioned above is quiescent and that the viscoelastic cylinder's surface is traction-free and exposed to a time-

dependent thermal shock. Then, the requirements for the boundary of the problem are as follows:

$$\theta(r, t) = \theta_0 H(t) \quad \text{at} \ r = a,$$
$$\sigma_{rr}(r, t) = 0 \qquad \text{at} \ r = a. \tag{19}$$

where $\theta_0$ is a constant. The following initial conditions are assumed:

$$|u(r, t)|_{t=0} = 0 = |\frac{\partial u(r, t)}{\partial t}|_{t=0},$$
$$|\theta(r, t)|_{t=0} = 0 = |\frac{\partial \theta(r, t)}{\partial t}|_{t=0}. \tag{20}$$

## 5 Temperature-dependent thermal properties

In contrast to previous research, the present work analyzes the temperature-dependent thermophysical properties and, consequently, the nature and behaviour of stress-induced thermal disturbances. The temperature-dependent properties of materials significantly influence thermal stress behaviour at high temperatures and high-temperature gradients. In the current investigation, the thermal conductivity $K$ and thermal rate $K^*$ as well as the specific heat coefficient $C_E$, are all assumed to be directly proportional to temperature [60, 61]

$$\{K, K^*, C_E\} = \{K_0, K_0^*, C_{E0}\}(1 + K_1\theta). \tag{21}$$

The thermal diffusion coefficient $k$, $(k = K/(\rho C_E))$ is assumed to be fixed in this case. The parameters $C_{E0}$, $K_0$ and $K_0^*$ are specific heat, thermal conductivity, and thermal rate, respectively, at room temperature $T_0$. The parameters represent the slope of the thermal conductivity/rate-temperature curves $K_1$ and $K_2$, which are also known as the slopes of the thermal conductivity/rate-temperature curves split by the intercepts $K_0$ and $K_0^*$.

By plugging Eq (21) into Eq (15), we get the following nonlinear partial differential equation

$$(1 + \tau_0 D_t^{(\alpha)}) \left[ \frac{\partial}{\partial t}\left(\rho C_E \frac{\partial \theta}{\partial t}\right) + T_0 \tau_m \frac{\partial^2}{\partial t^2}\left(\beta_{11}\frac{\partial u}{\partial r} + \beta_{22}\frac{u}{r}\right) \right] = \frac{\partial}{\partial t}(\nabla.((1 + K_1\theta)\nabla\theta)) + K_0^*(\nabla.((1 + K_1\theta)\nabla\theta)). \tag{22}$$

The previous equation can be converted into a linear equation by defining the following mapping [62]:

$$\psi = \int_0^\theta (1 + K_1\varphi)\mathrm{d}\varphi, \tag{23}$$

After inserting Eq (21) into Eq (23) and integrating Eq (23) we obtain

$$\psi = \theta\left(1 + \frac{1}{2}K_1\theta\right). \tag{24}$$

By differentiating the relation (23) twice, once in terms of distance and once in terms of time, the following equations may be deduced

$$(1 + K_1\theta)\nabla\theta = \nabla\psi, \quad \frac{K}{K_0}\frac{\partial \theta}{\partial t} = \frac{\partial \psi}{\partial t}. \tag{25}$$

When we differentiate Eq (25) in terms of distances, we obtain

$$\nabla.((1 + K_1\theta)\nabla\theta) = \nabla^2\psi, \tag{26}$$

where $\nabla^2 = \frac{\partial^2}{\partial r^2} + \frac{1}{r}\frac{\partial}{\partial r}$.

As a result, it is possible to use Eqs (25) and (26) to simplify the MGT heat transfer Eq (22) as

$$(1 + \tau_0 D_t^{(\alpha)})\left[\frac{1}{k}\frac{\partial\psi_1}{\partial t} + \frac{T_0}{K_0}\tau_m\frac{\partial^2}{\partial t^2}\left(\beta_{11}\frac{\partial u}{\partial r} + \beta_{22}\frac{u}{r}\right)\right] = \left(\frac{\partial}{\partial t} + \frac{K_0^*}{K_0}\right)\nabla^2\psi. \tag{27}$$

After employing Eq (26), the equation of motion (18) will take the following form

$$\tau_m\left[c_{11}\left(\frac{\partial^2 u}{\partial r^2} + \frac{1}{r}\frac{\partial u}{\partial r}\right) - c_{22}\frac{u}{r^2}\right]$$
$$= \frac{\beta_{11}}{1 + K_1\theta}\tau_m\frac{\partial\psi}{\partial r} + \frac{(\beta_{11} - \beta_{22})}{K_1 r}\tau_m\left(-1 + \sqrt{1 + 2K_1\psi}\right) + \rho\frac{\partial^2 u}{\partial t^2} - \rho\Omega^2 u. \tag{28}$$

The reference temperature $T_0$ is set so that condition $|\theta/T_0|"1$ is satisfied over the whole region. Therefore, Eq (28) becomes

$$\tau_m\left[c_{11}\left(\frac{\partial^2 u}{\partial r^2} + \frac{1}{r}\frac{\partial u}{\partial r}\right) - c_{22}\frac{u}{r^2}\right] = \frac{\beta_{11}}{1 + K_1\theta}\tau_m\frac{\partial\psi}{\partial r} + (\beta_{11} - \beta_{22})\tau_m\frac{\psi}{r} + \rho\frac{\partial^2 u}{\partial t^2} - \rho\Omega^2 u, \tag{29}$$

$$\begin{bmatrix} \sigma_{rr} \\ \sigma_{\xi\xi} \\ \sigma_{zz} \end{bmatrix} = \tau_m \begin{bmatrix} c_{11} & c_{12} & -\beta_{11} \\ c_{12} & c_{22} & -\beta_{22} \\ c_{13} & c_{23} & -\beta_{33} \end{bmatrix} \begin{bmatrix} \dfrac{\partial u}{\partial r} \\ \dfrac{u}{r} \\ \psi \end{bmatrix}. \tag{30}$$

Non-dimensional values listed below are included for simplicity

$$u', r', R'\} = \frac{c_0}{k}u, r, R\}, \quad t', \tau_0'\} = \frac{c_0^2}{k}t, \tau_0\}, \psi' = \frac{\psi}{T_0}, \sigma_{ij}' = \frac{\sigma_{ij}}{c_{11}}, \quad K_1' = T_0 K_1, \quad \Omega' = \frac{k}{c_0^2}\Omega, \quad c_0^2$$
$$= \frac{c_{11}}{\rho}. \tag{31}$$

Using the variables provided in (31), we get after removing dashes

$$\tau_m\left[\left(\frac{\partial^2 u}{\partial r^2} + \frac{1}{r}\frac{\partial u}{\partial r}\right) - c_2\frac{u}{r^2}\right] = \varepsilon_1\tau_m\frac{\partial\psi}{\partial r} + \varepsilon_0\tau_m\frac{\psi}{r} + \frac{\partial^2 u}{\partial t^2} - \Omega^2 u, \tag{32}$$

$$(1 + \tau_0 D_t^{(\alpha)})\left[\frac{\partial^2\psi_1}{\partial t^2} + \tau_m\frac{\partial^2}{\partial t^2}\left(\varepsilon_4\frac{\partial u}{\partial r} + \varepsilon_5\frac{u}{r}\right)\right] = \left(\frac{\partial}{\partial t} + \omega\right)\nabla^2\psi, \tag{33}$$

$$\begin{bmatrix} \sigma_{rr} \\ \sigma_{\xi\xi} \\ \sigma_{zz} \end{bmatrix} = \tau_m \begin{bmatrix} 1 & c_1 & -\varepsilon_1 \\ c_1 & c_2 & -\varepsilon_2 \\ c_3 & c_4 & -\varepsilon_3 \end{bmatrix} \begin{bmatrix} \dfrac{\partial u}{\partial r} \\ \dfrac{u}{r} \\ \psi \end{bmatrix}, \tag{34}$$

where

$$c_1 = \frac{c_{12}}{c_{11}}, \; c_2 = \frac{c_{22}}{c_{11}}, \; c_1 = \frac{c_{13}}{c_{11}}, \; c_1 = \frac{c_{23}}{c_{11}}, \; \varepsilon_1 = \frac{T_0 \beta_{11}}{c_{11}}, \; \varepsilon_2 = \frac{T_0 \beta_{22}}{c_{11}},$$

$$\varepsilon_3 = \frac{T_0 \beta_{33}}{c_{11}}, \; \varepsilon_4 = \frac{\beta_{11}}{\rho C_E}, \; \varepsilon_5 = \frac{\beta_{22}}{\rho C_E}, \; \varepsilon_0 = \frac{T_0 (\beta_{11} - \beta_{22})}{c_{11}}, \; \omega = \frac{K_0^*}{c_0^2 K_0}. \tag{35}$$

## 6 Analytical solutions

The governing equations comprise two independent variables, one spatial coordinate variable $r$, and a temporal variable $t$. The time function is also included in the boundary conditions (18). As a result, it is difficult to develop concrete solutions to the problem in the physical domain. The Laplace transform of the governing Eqs (32)–(34) concerning the instance time $t$ is combined with the initial conditions to provide the following equation in the Laplace domain (20)

$$\left( \frac{d^2 \bar{u}}{dr^2} + \frac{1}{r} \frac{d\bar{u}}{dr} \right) - \frac{\bar{u}}{r^2} - \frac{(s^2 - \Omega^2)}{\bar{\tau}_m} \bar{u} = \varepsilon_1 \frac{d\psi}{dr}, \quad \bar{\tau}_m = 1 + \eta s, \tag{36}$$

$$\left( 1 + \frac{s^\alpha \tau_0}{s^\alpha (1 - \alpha) + \alpha} \right) \left[ s^2 \psi + \varepsilon s^2 \bar{\tau}_m \left( \frac{d\bar{u}}{dr} + \frac{\bar{u}}{r} \right) \right] = (s + \omega) \nabla^2 \bar{\psi}, \tag{37}$$

$$\begin{bmatrix} \bar{\sigma}_{rr} \\ \bar{\sigma}_{\xi\xi} \\ \bar{\sigma}_{zz} \end{bmatrix} = \bar{\tau}_m \begin{bmatrix} 1 & c_1 & -\varepsilon_1 \\ c_1 & c_2 & -\varepsilon_2 \\ c_3 & c_4 & -\varepsilon_3 \end{bmatrix} \begin{bmatrix} \dfrac{d\bar{u}}{dr} \\ \dfrac{\bar{u}}{r} \\ \bar{\psi} \end{bmatrix}. \tag{38}$$

In the above equations, it is assumed $c_{11} = c_{22}$ and $\beta_{11} = \beta_{22}$. The overbar represents the Laplace transform of the appropriate function, while $s$ represents the Laplace variable. Eqs (36) and (37) have the following expressions:

$$\left( DD_1 - \frac{(s^2 - \Omega^2)}{\bar{\tau}_m} \right) \bar{u} = \varepsilon_1 \frac{d\bar{\psi}}{dr}, \tag{39}$$

$$\varepsilon q \bar{\tau}_m D_1 \bar{u} = (D_1 D - q) \bar{\psi}, \tag{40}$$

where

$$D = \frac{d}{dr}, \quad D_1 = \frac{d\bar{u}}{dr} + \frac{\bar{u}}{r}, q = \frac{s^2}{(s + \omega)} \left( 1 + \frac{s^\alpha \tau_0}{s^\alpha (1 - \alpha) + \alpha} \right). \tag{41}$$

The thermoelastic potential function $\phi$ is now introduced, which is described by the following relation

$$u = \frac{d\phi}{dr}. \tag{42}$$

Therefore, Eqs (39) and (40) are written as follows:

$$\left( D_1 D - \frac{(s^2 - \Omega^2)}{\bar{\tau}_m} \right) \bar{\phi} = \varepsilon_1 \bar{\psi}, \tag{43}$$

$$\varepsilon q \bar{\tau}_m D_1 D \bar{\phi} = (D_1 D - q) \bar{\psi}. \tag{44}$$

By removing $\bar{\psi}$ from Eqs (43) and (44), it is possible to get the following:

$$\left( (D_1 D)^2 - \left( q + \frac{(s^2 - \Omega^2)}{\bar{\tau}_m} + \varepsilon q \bar{\tau}_m \varepsilon_1 \right) (D_1 D) + \frac{q(s^2 - \Omega^2)}{\bar{\tau}_m} \right) \bar{\phi} = 0. \tag{45}$$

Eq (45) may be rewritten as:

$$(\nabla^2 - m_1^2)(\nabla^2 - m_2^2) \bar{\phi} = 0, \tag{46}$$

where $m_1^2$ and $m_2^2$ are the roots of the equation

$$m^2 - \left( q + \frac{(s^2 - \Omega^2)}{\bar{\tau}_m} + \varepsilon q \bar{\tau}_m \varepsilon_1 \right) m + \frac{q(s^2 - \Omega^2)}{\bar{\tau}_m} = 0. \tag{47}$$

The solution to Eq (46) under regularity conditions can be determined as follows

$$\bar{\phi} = \sum_{i=1}^{2} A_i K_0(m_i r), \tag{48}$$

where $K_0(m_i r)$ is a zero-order modified Bessel function of the second kind. $A_i$, $i = 1,2$ are constants that are independent of $r$. Using Eqs (43) and (48), the following solution of $\bar{\psi}$ may be found:

$$\bar{\psi} = \frac{1}{\varepsilon_1} \sum_{i=1}^{2} \left( m_i^2 - \frac{(s^2 - \Omega^2)}{\bar{\tau}_m} \right) A_i K_0(m_i r). \tag{49}$$

By combining Eq (48) with the Laplace transform of Eq (42), we can get

$$\bar{u} = - \sum_{i=1}^{2} m_i A_i K_1(m_i r). \tag{50}$$

With solutions to displacement $\bar{u}$ and the function $\bar{\psi}$, the following relationship can be used to calculate thermal stresses

$$x K_n'(x) = -x K_{n+1}(x) + n K_n(x). \tag{51}$$

The components of the stresses $\bar{\sigma}_{rr}, \bar{\sigma}_{\xi\xi}$ and $\bar{\sigma}_{zz}$ in the Laplace transform field are given in the following forms

$$\bar{\sigma}_{rr} = \sum_{i=1}^{2} \left[ \frac{(s^2 - \Omega^2)}{\bar{\tau}_m} A_i K_0(m_i r) + \frac{m_i(1 - c_1)}{r} A_i K_1(m_i r) \right], \tag{52}$$

$$\bar{\sigma}_{\xi\xi} = \sum_{i=1}^{2} \left[ \left( m_i^2(c_1 - 1) + \frac{(s^2 - \Omega^2)}{\bar{\tau}_m} \right) A_i K_0(m_i r) - \frac{m_i(c_1 + 1)}{r} A_i K_1(m_i r) \right], \tag{53}$$

$$\bar{\sigma}_{zz} = \sum_{i=1}^{2} \left[ \left( m_i^2 \left( c_3 - \frac{\varepsilon_3}{\varepsilon_1} \right) + \frac{\varepsilon_3}{\varepsilon_1} \frac{(s^2 - \Omega^2)}{\bar{\tau}_m} \right) A_i K_0(m_i r) - \frac{m_i(c_3 + c_4)}{r} A_i K_1(m_i r) \right]. \tag{54}$$

Eq (24) may be used to rewrite the boundary condition (19) as

$$\psi(r,t) = \theta_0 H(t) + \frac{K_1}{2}\left(\theta_0 H(t)\right)^2 \ \text{at} \ \ r = a. \tag{55}$$

We can get the following results after applying the Laplace transform to the boundary conditions (19) and (56)

$$\bar{\psi}(R,s) = \frac{\theta_0}{s} + \frac{K_1 \theta_0^2}{2s} = \bar{G}(s),$$
$$\bar{\sigma}_{rr}(R,s) = 0. \tag{56}$$

So, we have the following equations in the unknown parameters $A_i$, $i$ = 1,2:

$$\sum_{i=1}^{2}\left(m_i^2 - \frac{(s^2 - \Omega^2)}{\bar{\tau}_m}\right)A_i K_0(m_i R) = \varepsilon_1 \bar{G}(s), \tag{57}$$

$$\sum_{i=1}^{2}\left[\frac{(s^2 - \Omega^2)}{\bar{\tau}_m}A_i K_0(m_i R) + \frac{m_i(1 - c_1)}{R}A_i K_1(m_i R)\right] = 0. \tag{58}$$

Finally, the temperature $\bar{\theta}$ solution can be obtained by solving Eq (24) and applying the Laplace transform as follows:

$$\bar{\theta} = \frac{\sqrt{2k_1\bar{\psi} + 1} - 1}{K_1}. \tag{59}$$

Here the problem is solved analytically in the field of Laplace transform, and the next step is to find the inverse transformations of these fields.

## 7 Numerical inversion method

A variety of applications demonstrates the importance of numerical Laplace inversion. In engineering, Laplace transformation methods are frequently used to solve differential and integral equations and help in the application of other computing methods. Several numerical inversion techniques have been developed [63–66]. Numerous tests have shown that these processes are both easy and accurate.

This section identifies the inverse forms of field variables such as temperature, displacement, and thermal stresses within a rotating viscoelastic medium. To reverse the Laplace transform, we will use a numerical inversion method based on Fourier series expansion [67]. The effectiveness of the algorithm was verified by numerical testing. The following relationship [67] can be used to invert any Laplace domain function to the time domain:

$$f(r,t) = \frac{e^{\beta t}}{t}\left(\frac{1}{2}\bar{f}(r,\beta) + Re\sum_{n=1}^{m}(-1)^n \bar{f}\left(r, \beta + \frac{in\pi}{t}\right)\right). \tag{60}$$

The parameters $\beta$ and $m$ must be fine-tuned for better accuracy. The values of $\beta t$ are assumed to be from 4 to 5 were suggested by Mashayekhizadeh et al. [68]. After several tests, it was established that $\beta t$ = 4 and $m$ = 100 would produce much better results than any other values, even if the result is less sensitive to $m$ value when it exceeds certain thresholds.

## 8 Numerical example and discussion

This section discusses the rotational dependence, temperature-dependent characteristics, and fractional operators of rotating viscoelastic materials. Numerical calculations of a cobalt-like substance with a cylindrical hole were performed using Mathematica software programming. The mechanical and thermal characteristics of a cobalt-like material are described as [69]

$$\{c_{11}, c_{12}, c_{22}, c_{13}, c_{23}\} = \{3.071, 1.650, 1.027, 1.150, 3.581\} \times 10^{11} \text{kg m}^{-1}\text{s}^{-2},$$
$$\{\beta_{11} = \beta_{22}, \beta_{33}\} = \{7.04, 6.90\} \times 10^{6} \text{kg m}^{-2}\text{s}^{-2}, \ K_0 = 96 \text{ W m}^{-1}\text{K}^{-1},$$
$$\rho = 8836 \text{ kg m}^{-3}, \ T_0 = 298 \text{ K}, K_0^* = 2\text{W m}^{-1}\text{K}^{-1}\text{s}^{-1}, \theta_0 = 1.$$

### 8.1 Validation of results

It is important to first validate the mathematical models described in previous sections for estimating the thermoelastic behaviour of a viscoelastic rotating orthotropic hollow cylinder. Some of the available results provided by Aboueregal and Sedighi [24] are adopted for comparison with the existing solutions in Table 1 but in the absence of fractional order differentiation.

When comparing these results with the results obtained from literary works [24], it was discovered that there is a high degree of agreement in the behaviour of thermal and mechanical waves with variance in size. The existence of fractional derivatives operators decreases and dilates the response of thermomechanical waves, according to the data presented in Table 1. The current results approximate and agree well with those in reference [24], which indicates the validity of our model.

Within the limitations of this description, the effect of dimensionless physical field factors such as viscosity, rotation, and fractional parameters on thermoelastic interactions has been investigated. The numerical results are shown in Figs 1–12 for comparison and validation. Only numerical calculations will be carried out in the following three scenarios:

### 8.2 The effects of the temperature-dependent properties

Due to high temperatures, high gradient temperatures, and periodic temperature variations, thermal loads are imposed on structural components and mechanical elements in nuclear reactors, chemical plants, and high-speed aeroplanes, among other places. The properties of materials are affected by temperature, and the degree of dependency increases as the temperature rises. Fibre-reinforced composites and functionally gradient materials, for example, are

**Table 1. Comparison of the distribution temperature $\theta$ and radial stress $\sigma_{rr}$ with Ref. [24].**

| $r$ | Temperature $\theta$ | | Radial stress $\sigma_{rr}$ | |
|---|---|---|---|---|
| | Present | Ref. [24] | Present | Ref. [24] |
| 1 | 0.893049 | 0.893076 | 0 | 0 |
| 1.1 | 0.718442 | 0.751558 | -0.0778047 | -0.079634 |
| 1.2 | 0.713525 | 0.773075 | -0.3335000 | -0.294070 |
| 1.3 | 0.0602608 | 0.0360737 | -0.0610172 | 0.00432542 |
| 1.4 | 0.0357812 | 0.0372465 | -0.0166608 | -0.0182987 |
| 1.5 | 0.0334371 | 0.0364878 | -0.0167672 | -0.0185080 |
| 1.6 | 0.0285451 | 0.0323925 | -0.0146751 | -0.0168537 |
| 1.7 | 0.0195868 | 0.00454408 | -0.0107545 | -0.00500403 |
| 1.8 | 0.00478758 | 0.00172743 | -0.00375261 | -0.000915332 |
| 1.9 | 0.00145614 | 0.00164197 | -0.000764493 | -0.000862107 |
| 2 | 0.00125473 | 0.00145264 | -0.000655825 | -0.000770182 |

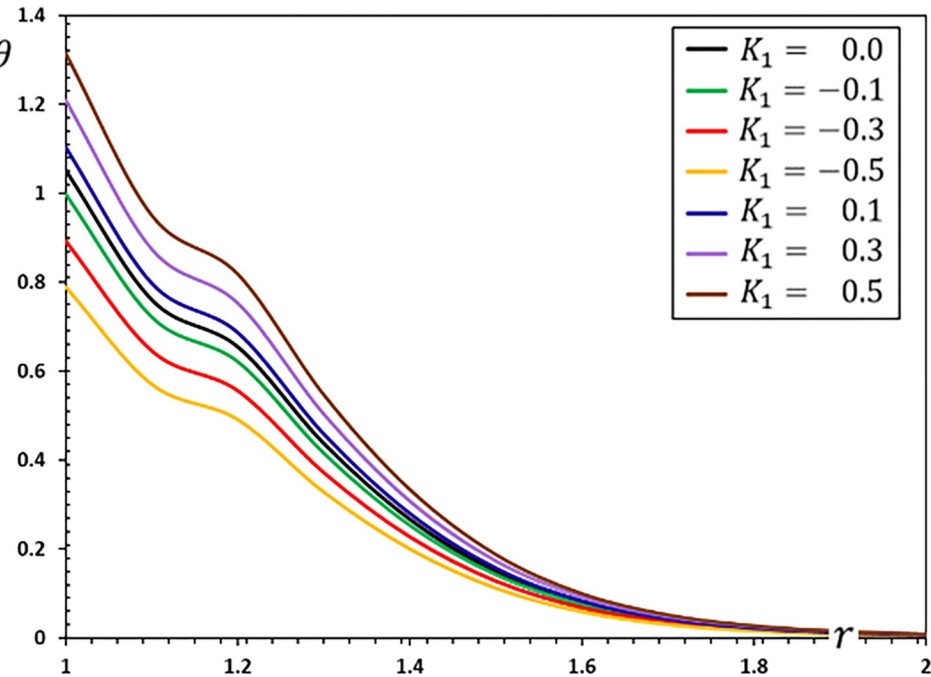

**Fig 1. The temperature variation $\theta$ for different values of the parameter $K_1$.**

becoming increasingly popular. As a result, more detailed thermomechanical behaviour evaluations of the structures mentioned above, components, and materials are required.

The fluctuations of the investigated non-dimensional field variables using different values of the coefficient of variation $K_1$ for the properties of refractory materials are studied in this section. To study and analyze the effect of the coefficient of variation $K_1$, six different values were selected. When the coefficient of thermal conductivity is variable and depends on temperature changes, we use the values $K_1 = 0.5, 0.3, 0.1, -0.1, -0.3, -0.5$, and when they are constant and independent of temperature changes, we use $K_1 = 0$.

The Moore–Gibson–Thompson thermoelastic (MGTE) heat conduction theory is investigated in this scenario, utilizing the fractional Atangana-Baleanu (AB) operator with nonsingular and nonlocal kernels. The angular speed $\Omega$, single relaxation time $\tau_0$, viscoelastic relaxation time and fractional-order measurements remain constant ($\Omega = 0.3$, $\eta = 0.01$, $\tau_0 = 0.2$ and $\alpha = 0.75$). Figs 1–4 depict comparisons and analyses of temperature $\theta$, radial displacement $u$, radial and hoop stresses $\sigma_{rr}$ and $\sigma_{\xi\xi}$ in accordance with the radial distance $r$. The numerical results given in the figures can be used to summarize the following fundamental findings:

- The coefficient of variation of temperature-dependent characteristics considerably influences all of the regions tested. It is evident that the nature of the variability between the field variables varies, and all thermoelastic models reveal that time has a major influence on all profiles.

- There is also the phenomenon of the restricted velocity of dissemination in all its forms. This is in contrast to the situation in which standard thermoelastic models have an infinite propagation speed, resulting in non-null values for all variables at every position in the unbounded solid.

- Temperature affects material properties, and the degree of dependency changes with temperature.

- Temperature-dependent material properties significantly influence thermal stress at high and gradient temperatures.

- The material deforms owing to thermal expansion caused by changing heat over time (thermal shock). The thermal expansion deformation of the evolution with radial material distances occurs when the size of the heat-disturbed area rises with time.

- The stress-free surface restricts deformation in the hollow, causing compressive thermal stresses in the solid and expansion due to varying heat. Thermal stresses grow in magnitude as time passes. However, this does not continue long due to the restricted dispersion of heat waves.

- The graph in Fig 1 shows how the temperature value decreases as the parameter $K_1$ is decreased. As seen in the graph, the temperature in a finite space domain has just one non-zero value at any given time. The disturbances dissipate, and the area is devoid of thermal turbulence. Only in specific places throughout time is the nonzero region transmitted properly.

- As seen in Fig 2, the absolute amount of displacement $u$ increases as the value climbs to the point where the curves connect. Following the intersection, the absolute amount of displacement decreases as the value of the parameter $K_1$ grows.

- Fig 3 depicts how the variations in radial thermal stress $\sigma_{rr}$ start at zero for all situations at surface cavity $r = a$ that are compatible with the restricted condition and gradually decline to their lowest value.

- As illustrated in Fig 3, the thermal conductivity change parameter $K_1$ decreases the amplitude of the stress $\sigma_{rr}$.

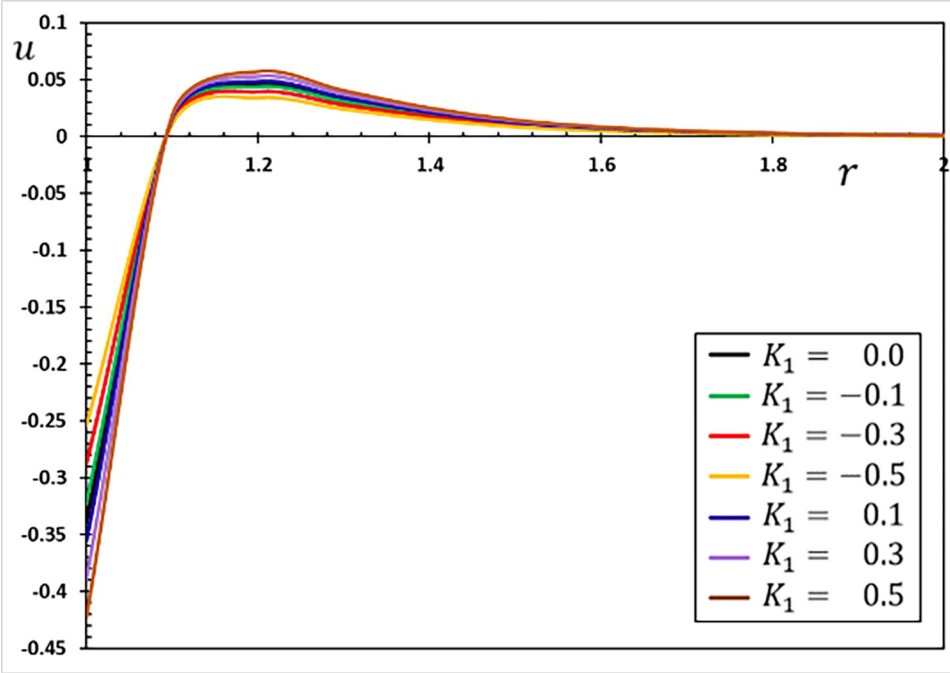

**Fig 2. The displacement variation $u$ for different values of the parameter $K_1$.**

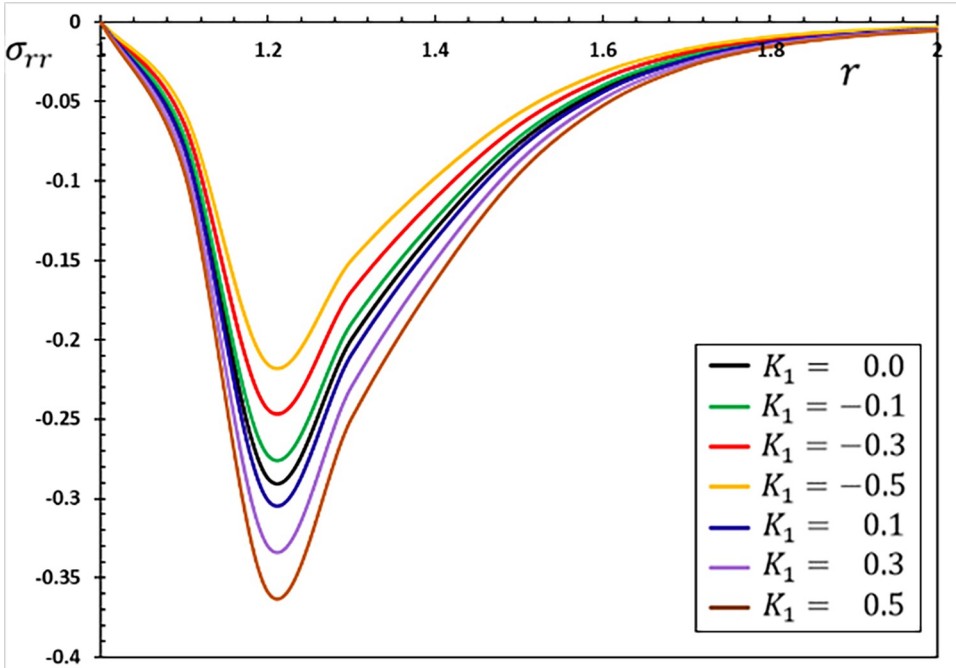

**Fig 3. The radial stress variation $\sigma_{rr}$ for different values of the parameter $K_1$.**

- Under all conditions, the hoop thermal stress begins negatively, climbs to its maximum at $r = 1.1$, rapidly decreases to a lower limit at $r = 1.2$, and gradually increases to zero and the steady-state (see Fig 4). The graph also shows that as the value of the indicator $K_1$ falls so does the hoop stress $\sigma_{\xi\xi}$.

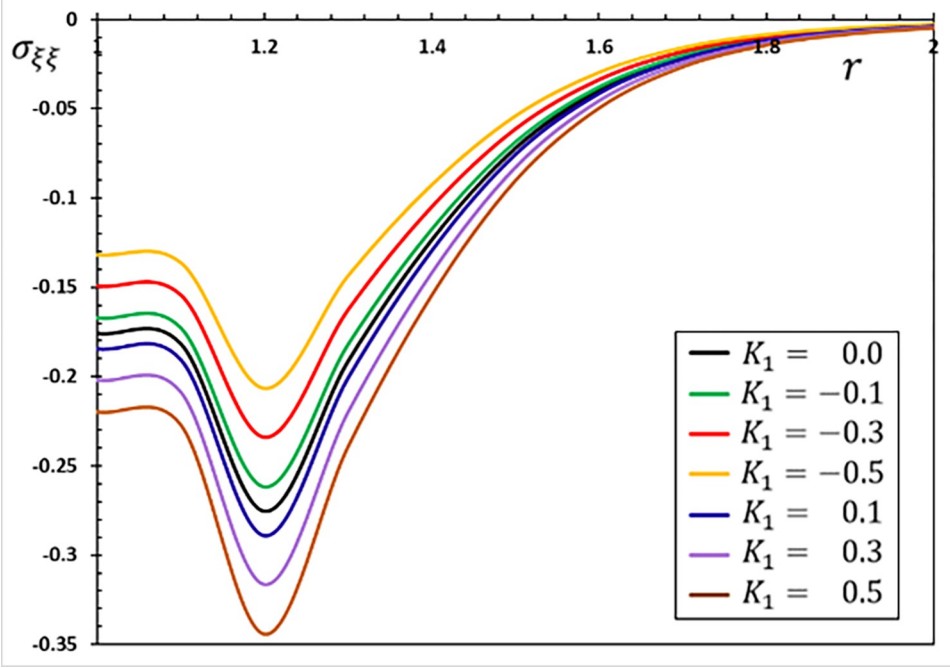

**Fig 4. The hoop stress variation $\sigma_{\xi\xi}$ for different values of the parameter $K_1$.**

- Tensile stress increases with time in the material adjacent to the cylinder surface. The greatest values of the studied fields appear more often on the cavity's surface, and their amplitude decreases as radial distances grow.

- The numerical findings reveal that the oscillatory thermal conductivity factor significantly impacts all physical fields, emphasizing the significance of considering this component and its temperature dependency. As a result, both engineering and production applications must account for these variations.

## 8.3 Influence of the angular velocity

Based on the Moore–Gibson–Thompson fractional thermo-viscoelastic model, the second scenario examines non-dimensional temperature, displacement, and thermal stress versus various angular velocity values (FV-MGTE). The fractional Atangana-Baleanu (AB) operator [35] is utilized, which has nonsingular and nonlocal kernels. In this situation, the thermal relaxation time $\tau_0$, the variation coefficient of thermal conductivity K 1, the fractional parameter $\alpha$, and the viscosity parameter $\eta$ are assumed to be fixed. ($K_1 = -0.3$, $\eta = 0.01$, $\tau_0 = 0.2$ and $\alpha = 0.75$). When the medium rotates, we take $\Omega = 1,3$; otherwise, we take $\Omega = 0$. The rotation parameter has a considerable impact on all fields (see Figs 5–8).

Fig 5 depicts the temperature variations $\theta$ as a function of radial distance $r$. At any one time, only a small section of the interior surface of the hollow cylinder may be discovered to be non-null. In all three scenarios, the temperature $\theta$ begins with its highest value near the cylinder's inner border. When compared to zero angular velocity $\Omega$ values, the amplitude of temperature $\theta$ fluctuations is minimal for large angular velocity values, indicating a growing impact rotation.

Fig 6 shows the relationship between displacement $u$ and radial distance $r$ for various angular velocity $\Omega$ values. The displacement $u$ seems to be highest near the cavity's boundary. Due

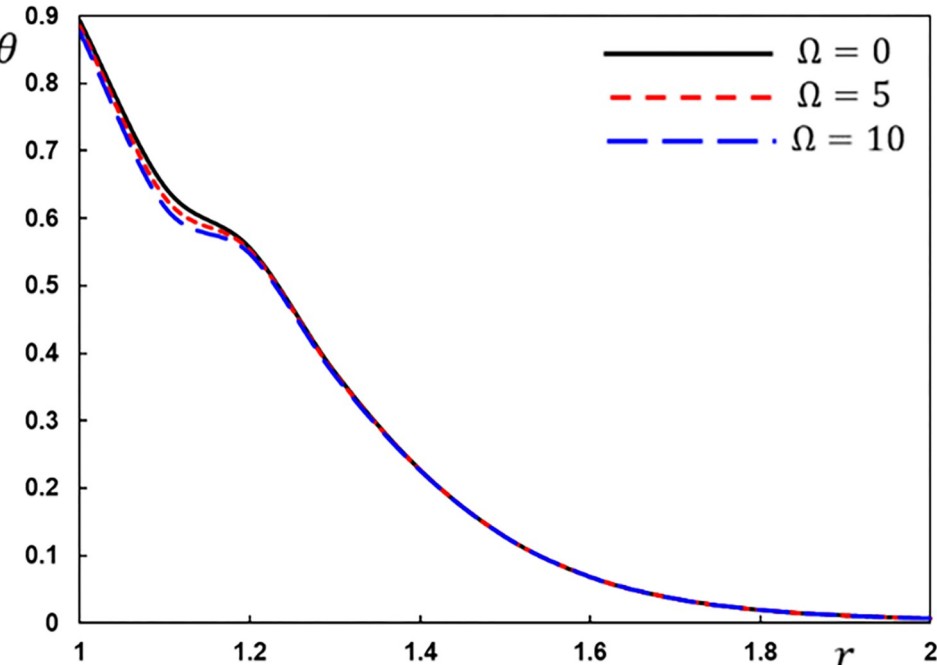

**Fig 5. The temperature variation $\theta$ for different values of the rotation speed $\Omega$.**

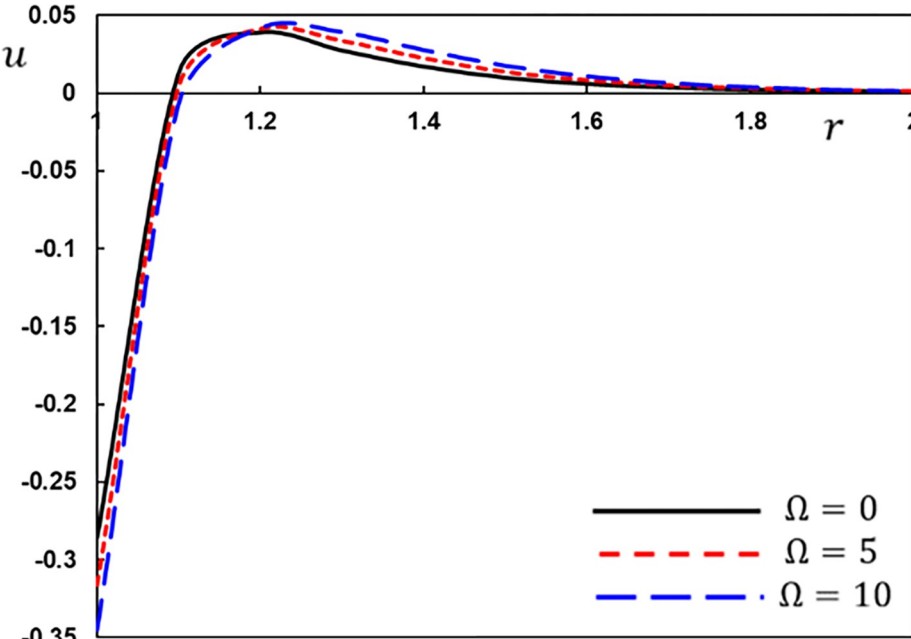

**Fig 6. The displacement variation $u$ for different values of the rotation speed $\Omega$.**

to the heat wave effect, the radial displacement region is always restricted to a nonzero area. The displacement field $u$ exhibits the same quality of behaviour in various magnitudes for every rotational velocity $\Omega$, as shown in Fig 6. The displacement $u$ reduces as the rotation parameter lowers before $r = 1.6$ and reverses the behaviour in others. Finally, as the radial distance r grows larger, the curves converge on zero values.

Fig 7 displays the evolution of the thermal stress $\sigma_{rr}$ versus the distance $r$ when $\Omega = 0,1,3$. Because of the presence of the rotational factor, the amplitude of the thermal stress $\sigma_{rr}$ is smaller than that of a nonrotating material. Fig 7 also shows the nonzero stress region, which means that the wave effect of heat is limited. The thermal stress $\sigma_{\xi\xi}$ starts with negative values in all situations, rises and then declines to the minimum value at $r = 1.2$, rapidly rises to the highest value at $r = 1.4$, and then progressively rises until it tends to zero and the steady-state. Fig 8 also demonstrates that as the rotational parameter $\Omega$ is reduced, the thermal stress $\sigma_{\xi\xi}$ rises.

## 8.4 Comparison of classical and modified fractional operators

A new approach for calculating the temporal fractional heat conduction equation is provided in the current research. The Atangana–Baleanu (AB) derivative, according to Caputo, is the fractional derivative operator in use. In the presented fractional thermal conductivity, the singular, nonlocal Mittag–Leffler function acts as a kernel. The modified heat equation replaces Fourier's law and the parabolic heat transfer equation with more general equations that account for the complex internal structure of the medium as well as microscopic physical processes.

The Moore–Gibson–Thompson fractional heat conduction model with the Atangana–Baleanu operator (FABMGTE) was compared to the fractional thermoelastic model with the standard Caputo operator in this section (FCMGTE). We will consider how the fractional characteristics of the Atangana-Baleanu (AB) and Caputo operators influence non-

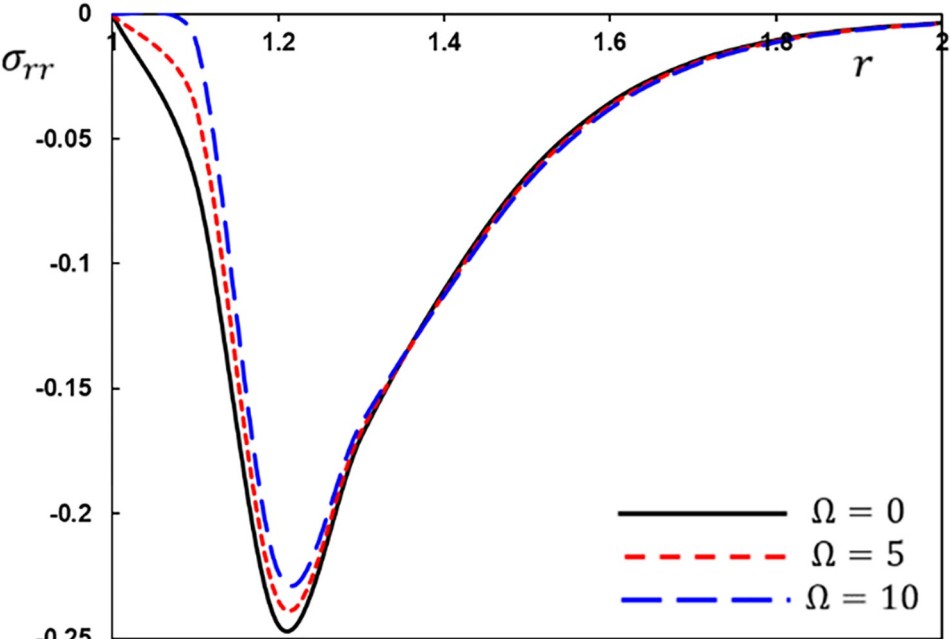

**Fig 7. The radial stress variation $\sigma_{rr}$ for different values of the rotation speed $\Omega$.**

dimensional temperature, displacement, and stresses. The Laplace transform of the conventional fractional Caputo operator in the case of a zero initial condition is as follows:

$$\mathcal{L}[D_t^{(\alpha)}f(t)] = s^\alpha \mathcal{L}[f(t)] - f(0) = s^\alpha F(s).$$

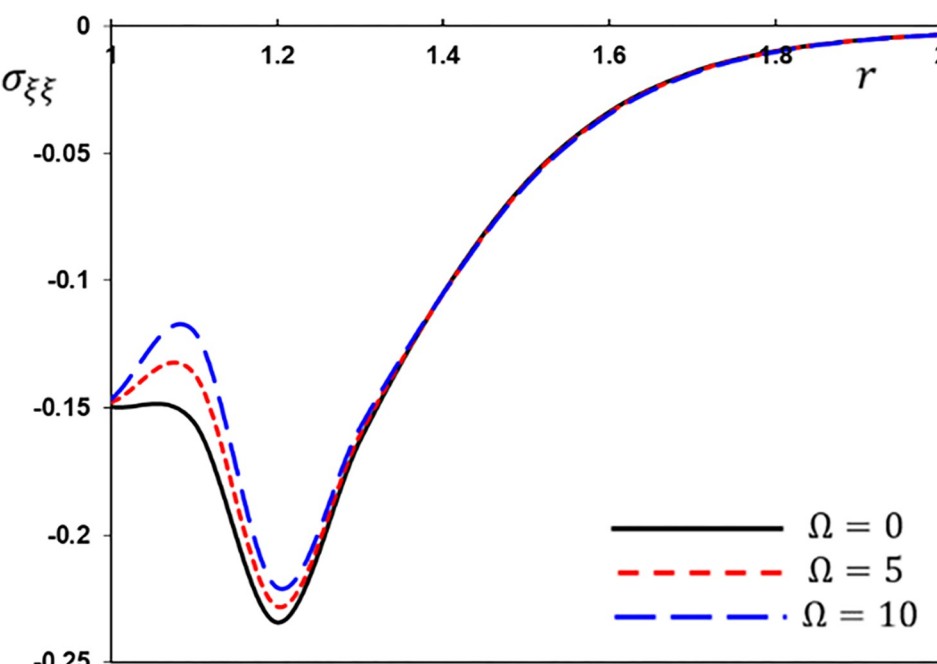

**Fig 8. The hoop stress variation $\sigma_{\xi\xi}$ for different values of the rotation speed $\Omega$.**

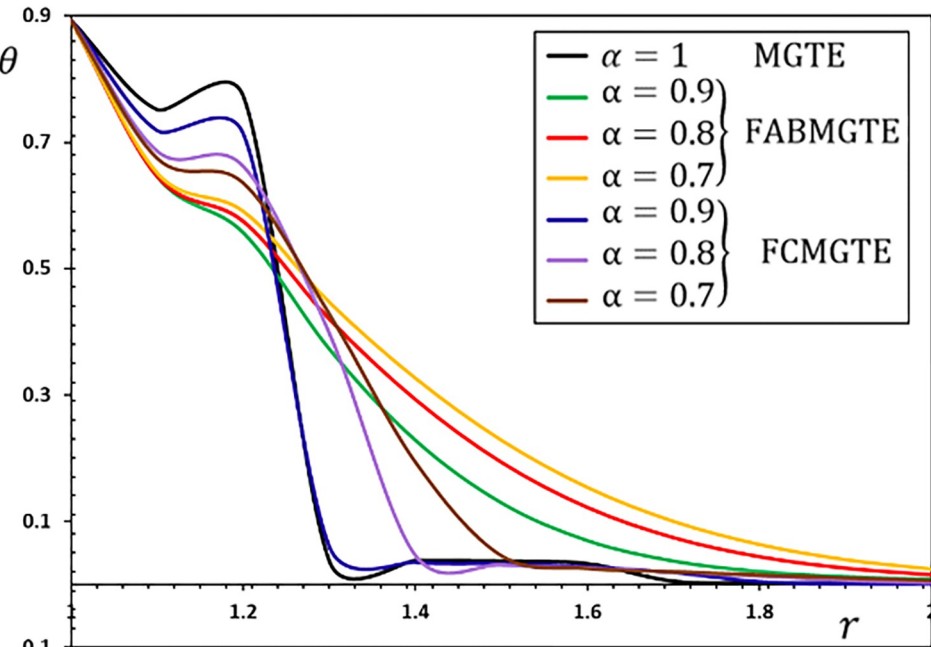

**Fig 9. The temperature variation $\theta$ for different fractional operator derivatives.**

Figs 9–12 illustrate the changes in the examined fields as a function of distance $r$. When the fractional-order parameters $\alpha = 0.9$, $\alpha = 0.8$ and $\alpha = 0.7$ are taken into consideration, the values $K_1 = -0.3$, $\eta = 0.01$, $\tau_0 = 0.2$ and $\Omega = 3$ are used in this analysis and numerical estimation. Furthermore, all calculation results for $\alpha = 1$ are quantitatively tested to ensure accuracy and to confirm another fractional check adjustment.

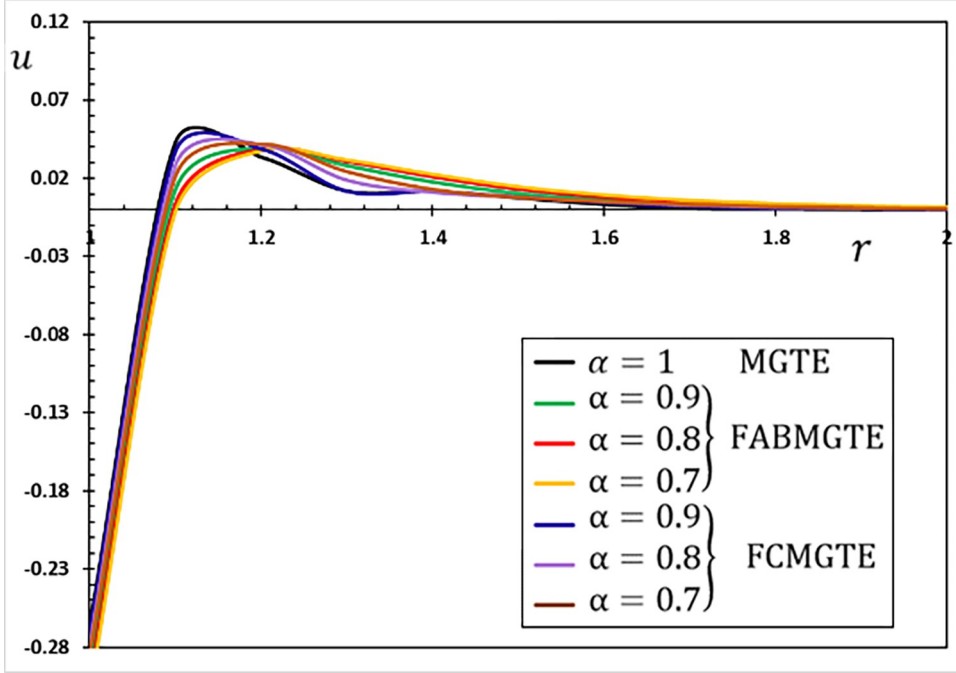

**Fig 10. The displacement variation $u$ for different fractional operator derivatives.**

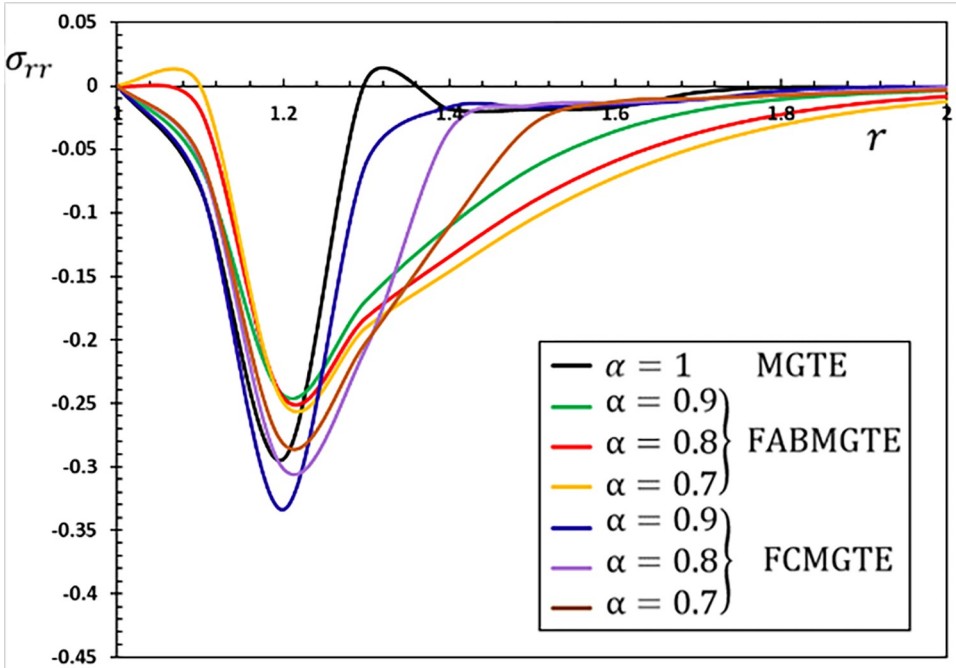

**Fig 11. The radial stress variation $\sigma_{rr}$ for different fractional operator derivatives.**

The following main conclusions can be drawn from the figures:

• The current model can degenerate into a classical version in the case of $\alpha = 1$.

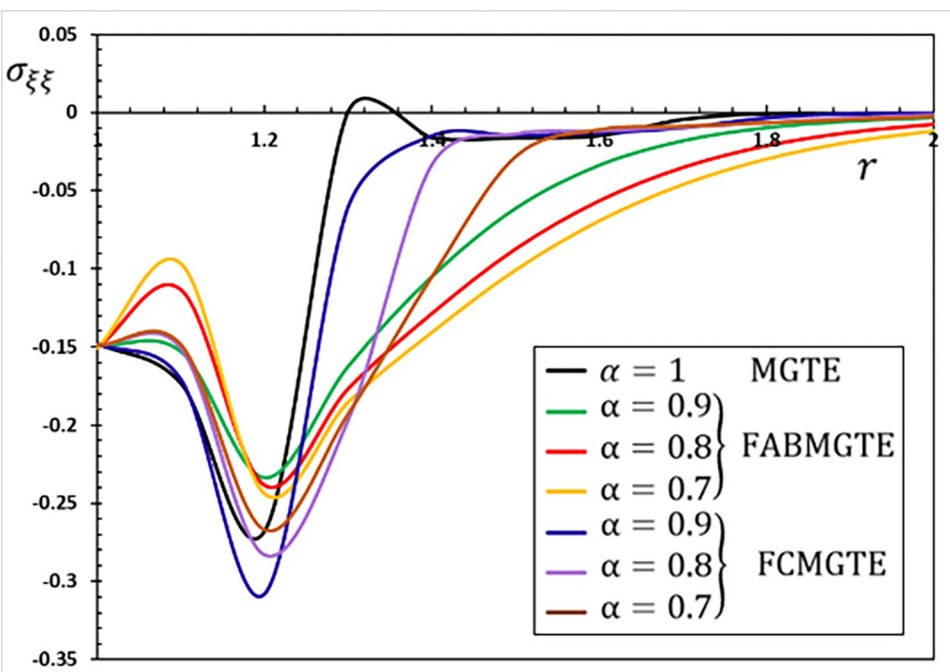

**Fig 12. The hoop stress variation $\sigma_{\xi\xi}$ for different fractional operator derivatives.**

- The findings show that fractional derivatives are required to reduce the size of the field profiles under consideration.

- The decay rate is faster in the modified fractional models than in the classical model. This phenomenon occurs in viscous and non-viscous materials.

- The maximum amplitude of the physical fields decreases with the decrease of the fractional-order factor.

- Fractional derivatives have a slight impact on the temperature distribution and may be completely absent

- We can classify materials using the innovative MGTE framework based on their fractional-order properties. As a result, the fractional-parameter $\alpha$ becomes more important as a measure of the heat transfer capacity of the conducting material.

- The Atangana–Baleanu operator tends to enhance the temperature and thermal stress patterns compared to Caputo operators, as seen in Figs 9, 11, and 12.

- The temperature distribution in the FABMGTE idea is much larger than in the MGTE model, and the AB fractional operator substantially influences the displacement.

- One of the most significant conclusions drawn from examining the various field profiles is that the thermal and mechanical wave action is smoother in the FABMGTE system than in the FCMGTE model and in the typical case without the fractional derivative.

- As shown in the above literature, fractional derivatives have several applications in mathematical modelling and the investigation of real-world phenomena.

- Because of its wide applications in biological, physical, and medical engineering and some other nonlinear studies, the recently developed Atangana–Baleanu fraction operator has gained attention and respect.

- Features showing the interaction of materials due to propagation of thermomechanical vibrations are more flexible in the Atangana-Baleanu fractional heat conduction model (FABMGTE) than in the Caputo fractional heat transfer (FCMGTE) system.

- The fractional variance effect of the Atangana–Baleanu fractional derivative operator is more realistic and adaptable than that of the Caputo derivative operator. It can be utilized to describe many real-world conditions with confidence.

- When using the Atangana–Baleanu fractional derivative operator, raising the fractional differential order reduces the value of the physical variables, causing different distributions to disappear more quickly.

## 8.5 The effect of the viscosity term

The last scenario examines how the temperature, displacement, and stresses in non-dimensional forms vary with viscoelastic relaxation $\eta$ owing to the viscosity factor $\tau_m = 1 + \eta \frac{\partial}{\partial t}$ in the system equations. The validity of the distributions in three distinct dimensionless values of mechanical relaxation time (viscosity) $\eta$ owing to viscosity was explored. In this scenario, the viscoelastic Moore–Gibson–Thompson thermoelastic (FV-MGTE) heat conduction theory is investigated based on the fractional Atangana-Baleanu (AB) operator with nonsingular and nonlocal kernels. The angular speed $\Omega$ values, the single relaxation time $\tau_0$, the viscoelastic

relaxation time $\eta$ and the fractional-order parameters remain fixed ($K_1 = -0.3$, $\Omega = 0.3$, $\eta = 0.01$, $\tau_0 = 0.2$ and $\alpha = 0.75$).

In two cases, comparisons between the dimensionless values of the studied fields were made. The first case is when the viscosity $\tau_m$ term ($\eta = 0.01$ and $\eta = 0.02$) is introduced, and the fractional viscoelastic MGTE model (FV-MGTE) is used. The second case is when the viscosity term $\tau_m$ is neglected ($\eta = 0.0$), and the fractional model for non-viscosity materials (F-MGTE) is applied. Figs 13–16 are displayed to investigate the influence of viscosity on thermophysical characteristics of FV-MGTE and F-MGTE models.

From the figures presented, the following important points can be deduced:

- Fig 13 depicts the influence of the viscosity parameter $\eta$ on temperature distribution.

- The mechanical viscosity parameter $\eta$ has been shown to have a significant influence on temperature variations.

- Compared to the F-MGTE theory, the temperature change in the FV-MGTE theory is much wider.

- The factor of viscosity $\eta$ significantly increases the temperature profile.

- Fig 14 shows comparisons between displacement $u$ and the viscosity parameter $\eta$.

- It is shown that the displacement $u$ decreases as the viscoelastic relaxation parameter $\eta$.

- Furthermore, the displacement variation of F-MGTE appears to be greater than that of FV-MGTE.

- The radial displacement in both models follows the same trend.

Figs 11 and 12 show how the absolute values of stresses $\sigma_{rr}$ and $\sigma_{\xi\xi}$ rise with the rise of viscosity parameter $\eta$. As time changes, the effect of viscosity within the body fades away from

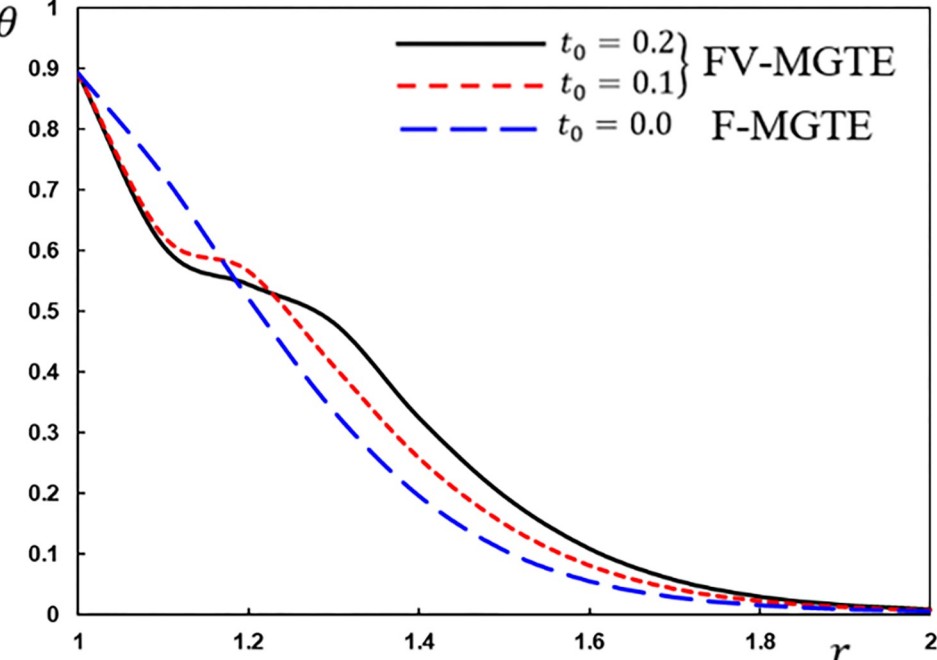

**Fig 13. The temperature variation $\theta$ for different viscosity parameter $\eta$.**

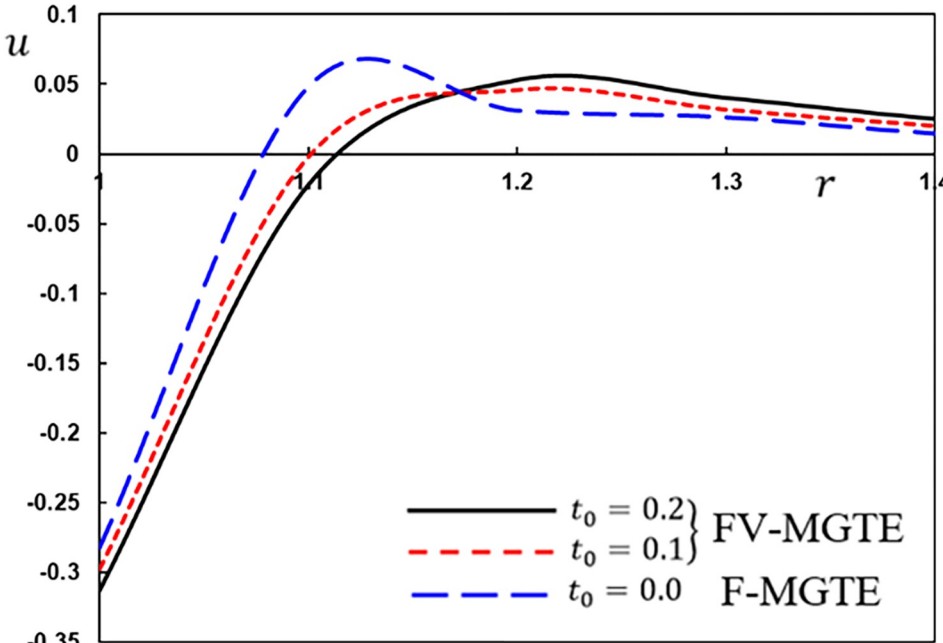

**Fig 14. The displacement variation $u$ for different viscosity parameter $\eta$.**

the inner surface of the cylinder. The stresses are very sensitive to the impact of viscosity. The presence of the viscosity factor reduces the amplitude of the stresses in both formulations. The results in this field will benefit researchers in materials science, material designers, low-temperature physicists, and those researching the hyperbolic viscosity theory of thermoelasticity.

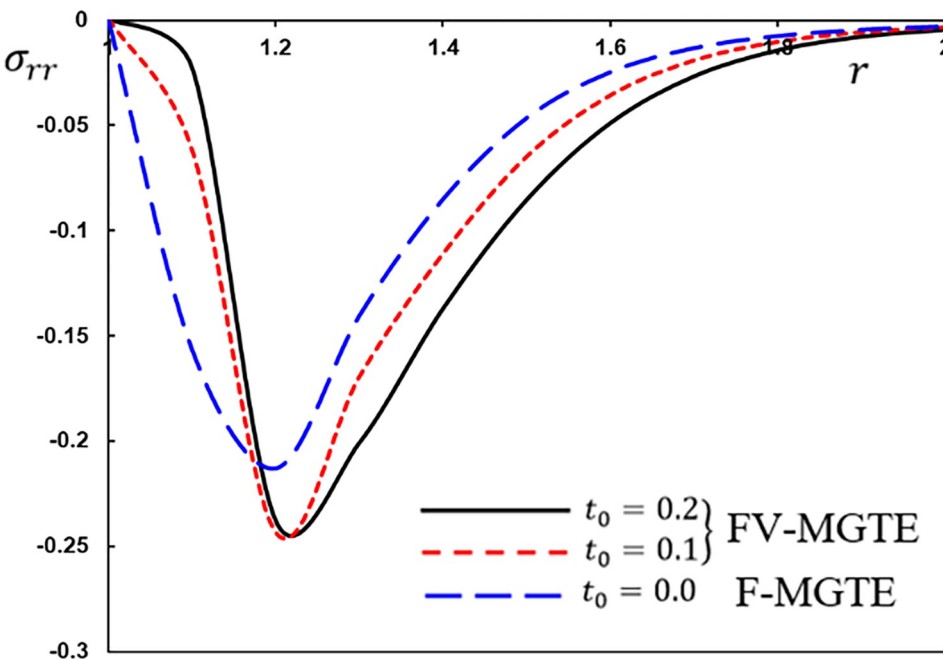

**Fig 15. The radial stress variation $\sigma_{rr}$ for different viscosity parameter $\eta$.**

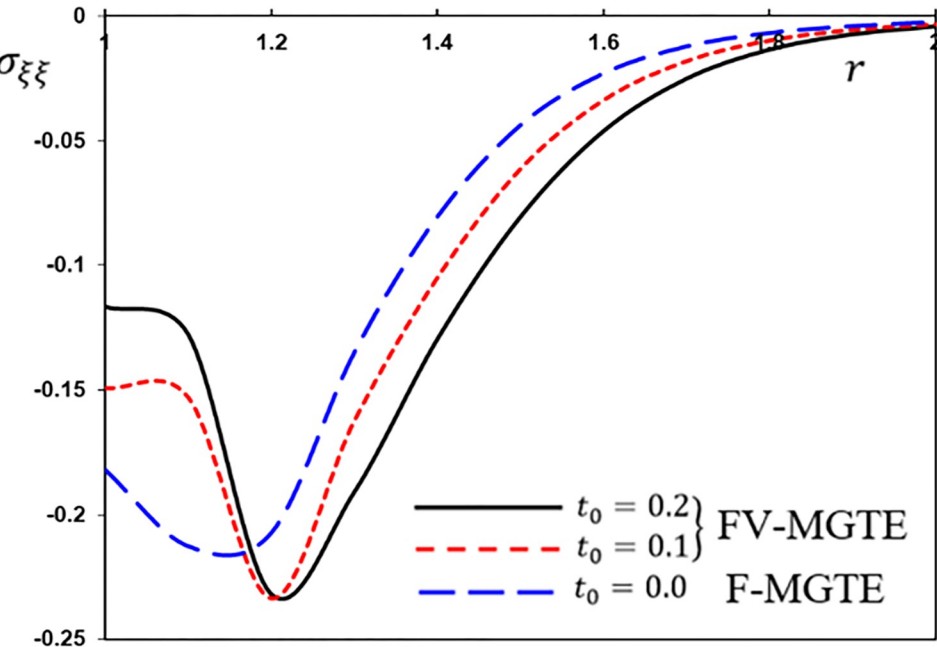

**Fig 16. The hoop stress variation $\sigma_{\xi\xi}$ for different viscosity parameter $\eta$.**

## 9 Concluding remarks

In this present analysis, a fractional mathematical model of thermo-viscoelastic heat transfer in the sense of Kelvin-Voigt type is proposed. The system of equations is based on the Moore-Gibson-Thompson heat equation, which includes the fractional Atangana-Baleanu (AB) operator. The problem is solved numerically using non-dimensional variables and the Laplace transform technique. The issue of thermoelasticity in one dimension of an infinitely rotating body with a spherical cavity is studied numerically, and the following can be said:

- The fractional variance effect of the Atangana–Baleanu fractional derivative operator is more realistic and adaptable than that of the Caputo derivative operator. It can be used to describe many real-world conditions confidently.

- Changing the thermal properties of a material, such as the coefficient of thermal conductivity and its dependence on temperature change, significantly affects its behaviour in different physical domains. As a result, these modifications must be considered in engineering and manufacturing applications.

- Due to their existence, derivatives of fractional orders have a major impact on the distribution of thermo-viscoelastic material field quantities. It appears that some properties of the thermophysical amount of matter change in volume due to the presence of fractional derivatives in the thermal conductivity equation.

- Due to the viscosity term, the size of the thermophysical field variables is reduced and the physical fields decay. As a result, the viscosity parameter has a prominent influence on the distributions of all studied thermophysical fields.

- New materials can be classified based on the Atangana-Baleanu fractional index, which may be the basis for using temperature-dependent thermo-viscous materials.

- These theoretical findings will be valuable to experimental scientists and researchers researching this area. In the heat flow of a flexible second-order viscous fluid and a Maxwell fluid, the fractional AB derivative can also be used to obtain experimental results.

## Author Contributions

**Conceptualization:** Ahmed E. Abouelregal, Meshari Alesemi.

**Data curation:** Ahmed E. Abouelregal.

**Formal analysis:** Ahmed E. Abouelregal, Meshari Alesemi.

**Investigation:** Meshari Alesemi.

**Methodology:** Meshari Alesemi.

**Project administration:** Ahmed E. Abouelregal.

**Software:** Ahmed E. Abouelregal.

**Supervision:** Meshari Alesemi.

**Validation:** Ahmed E. Abouelregal.

**Visualization:** Meshari Alesemi.

**Writing – original draft:** Ahmed E. Abouelregal.

**Writing – review & editing:** Ahmed E. Abouelregal, Meshari Alesemi.

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
