## [Decision Letter · Decision Letter 0]

1 Dec 2021

PONE-D-21-36348Fractional Moore-Gibson-Thompson heat transfer model with nonlocal and nonsingular kernels of a rotating viscoelastic annular cylinder with changeable thermal propertiesPLOS ONE

Dear Dr. Abouelregal,

Thank you for submitting your manuscript to PLOS ONE. After careful consideration, we feel that it has merit but does not fully meet PLOS ONE’s publication criteria as it currently stands. Therefore, we invite you to submit a revised version of the manuscript that addresses the points raised during the review process.

We look forward to receiving your revised manuscript.

Kind regards,

Mohammad Mehdi Rashidi

Academic Editor

PLOS ONE

Journal Requirements:

" ext-link-type="uri" xlink:type="simple">https://journals.plos.org/plosone/s/file?id=ba62/PLOSOne_formatting_sample_title_authors_affiliations.pdf"

The authors extend their appreciation to the Deanship of Scientific Research at Jouf University for funding this work through research grant No. (DSR-2021-03-0377). We would also like to extend our sincere thanks to the College of Science and Arts in Al-Qurayyat for its technical support.

Ahmed E Abouelregal

Jouf University 

DSR-2021-03-0377

Ahmed E Abouelregal

Jouf University 

DSR-2021-03-0377

5. Please amend the manuscript submission data (via Edit Submission) to include author Meshari Alesemi.

Reviewers' comments:

Reviewer's Responses to Questions

**Comments to the Author**

1. Is the manuscript technically sound, and do the data support the conclusions?

Reviewer #1: Yes

Reviewer #2: Yes

2. Has the statistical analysis been performed appropriately and rigorously? 

Reviewer #1: Yes

Reviewer #2: Yes

3. Have the authors made all data underlying the findings in their manuscript fully available?

Reviewer #1: Yes

Reviewer #2: Yes

4. Is the manuscript presented in an intelligible fashion and written in standard English?

Reviewer #1: Yes

Reviewer #2: Yes

5. Review Comments to the Author

Reviewer #1: In manuscript No. PONE-D-21-36348, a thermoelastic fractional heat conduction model was developed based on the Moore-Gibson-Thompson equation to examine the axial symmetry problem of a viscoelastic orthotropic hollow cylinder. Some new results are reported. However, the following comment should be considered,

- Please enhance the abstract with some quantitative results. For instance, it is mentioned that ‘The results showed that the effective variables have a significant effect on the responses to the variables of the studied fields.’, please mention what is a significant effect?

- Also, please provide important quantitative results in the conclusion.

- Please provide a ‘Nomenclature’.

- It seems that some figures can be combined. For example, the authors can join Figs. 13 to 16 to reduce the number of figures.

- The authors should provide the details of validation in Section 8.

- The literature review can be updated with some other studies on fluid flow and heat transfer inside tubes and channels. For instance, https://doi.org/10.1016/j.cep.2015.08.009 and https://doi.org/10.1016/j.applthermaleng.2016.07.173

- Finally, I would suggest the authors double-check all sentences for English or ask someone to help with the writing.

Reviewer #2: Review Report:

The present manuscript is very interesting, but it needs some minor revisions before its publication as follows:

1. The literature review about the Atangana-Baleanu fractional derivative is not complete. Please review recently published papers.

2. How do the authors validate their results?

3. Please check the comma and the point at the end of each expression. Please see Eq (19).

4. The conclusion should be written as the past.

5. The authors can study the following relevant papers and complete their references list: Mathematics and Computers in Simulation 187 (2021) 248-260 and Mathematical Methods in the Applied Sciences 44 (2021) 6247-6258.

After addressing the above suggestions, I recommend the current manuscript for publication.

6. PLOS authors have the option to publish the peer review history of their article (what does this mean?). If published, this will include your full peer review and any attached files.

Reviewer #1: No

Reviewer #2: No

---

## [Author Response · Author response to Decision Letter 0]

12 May 2022

The first author is the one who will receive the grant (DSR-2021-03-0385) from Al-Jouf University after publishing the research.

I received an email about this and I don't know what the exact problem is

---

## [Decision Letter · Decision Letter 1]

30 May 2022

Fractional Moore-Gibson-Thompson heat transfer model with nonlocal and nonsingular kernels of a rotating viscoelastic annular cylinder with changeable thermal properties

PONE-D-21-36348R1

Dear Dr. Abouelregal,

We’re pleased to inform you that your manuscript has been judged scientifically suitable for publication and will be formally accepted for publication once it meets all outstanding technical requirements.

Kind regards,

Mohammad Mehdi Rashidi

Academic Editor

PLOS ONE

Additional Editor Comments (optional):

Reviewers' comments:

Reviewer's Responses to Questions

**Comments to the Author**

1. If the authors have adequately addressed your comments raised in a previous round of review and you feel that this manuscript is now acceptable for publication, you may indicate that here to bypass the “Comments to the Author” section, enter your conflict of interest statement in the “Confidential to Editor” section, and submit your "Accept" recommendation.

Reviewer #1: All comments have been addressed

Reviewer #2: (No Response)

2. Is the manuscript technically sound, and do the data support the conclusions?

Reviewer #1: Yes

Reviewer #2: (No Response)

3. Has the statistical analysis been performed appropriately and rigorously? 

Reviewer #1: Yes

Reviewer #2: (No Response)

4. Have the authors made all data underlying the findings in their manuscript fully available?

Reviewer #1: Yes

Reviewer #2: (No Response)

5. Is the manuscript presented in an intelligible fashion and written in standard English?

Reviewer #1: Yes

Reviewer #2: (No Response)

6. Review Comments to the Author

Reviewer #1: All comments have been addressed by the authors. Now it can be accepted for publication. All comments have been addressed by the authors. Now it can be accepted for publication.

Reviewer #2: (No Response)

7. PLOS authors have the option to publish the peer review history of their article (what does this mean?). If published, this will include your full peer review and any attached files.

Reviewer #1: No

Reviewer #2: No

---

## [Editor Report · Acceptance letter]

8 Jun 2022

PONE-D-21-36348R1 

Fractional Moore-Gibson-Thompson heat transfer model with nonlocal and nonsingular kernels of a rotating viscoelastic annular cylinder with changeable thermal properties 

Dear Dr. Abouelregal:

I'm pleased to inform you that your manuscript has been deemed suitable for publication in PLOS ONE. Congratulations! Your manuscript is now with our production department. 

Kind regards, 

on behalf of

Professor Mohammad Mehdi Rashidi 

Academic Editor

PLOS ONE